# Evolution of DNA replication origin specification and gene silencing mechanisms

Y. Hu[1,2], A. Tareen[1,3], Y-J. Sheu[1], W. T. Ireland[4], C. Speck [5], H. Li [6], L. Joshua-Tor [1,7], J. B. Kinney [1,3] & B. Stillman [1✉]

DNA replication in eukaryotic cells initiates from replication origins that bind the Origin Recognition Complex (ORC). Origin establishment requires well-defined DNA sequence motifs in *Saccharomyces cerevisiae* and some other budding yeasts, but most eukaryotes lack sequence-specific origins. A 3.9 Å structure of *S. cerevisiae* ORC-Cdc6-Cdt1-Mcm2-7 (OCCM) bound to origin DNA revealed that a loop within Orc2 inserts into a DNA minor groove and an α-helix within Orc4 inserts into a DNA major groove. Using a massively parallel origin selection assay coupled with a custom mutual-information-based modeling approach, and a separate analysis of whole-genome replication profiling, here we show that the Orc4 α-helix contributes to the DNA sequence-specificity of origins in *S. cerevisiae* and Orc4 α-helix mutations change genome-wide origin firing patterns. The DNA sequence specificity of replication origins, mediated by the Orc4 α-helix, has co-evolved with the gain of ORC-Sir4-mediated gene silencing and the loss of RNA interference.

[1] Cold Spring Harbor Laboratory, 1 Bungtown Road, Cold Spring Harbor, NY 11724, USA. [2] Program in Molecular and Cell Biology, Stony Brook University, Stony Brook, NY 11794, USA. [3] Simons Center for Quantitative Biology, Cold Spring Harbor Laboratory, Cold Spring Harbor, NY 11724, USA. [4] Department of Physics, California Institute of Technology, Pasadena, CA 91125, USA. [5] DNA Replication Group, Institute of Clinical Sciences, Faculty of Medicine, Imperial College London, W12 0NN London, UK. [6] Structural Biology Program, Van Andel Institute, Grand Rapids, MI 49503, USA. [7] W. M. Keck Structural Biology Laboratory, Howard Hughes Medical Institute, Cold Spring Harbor, NY 11724, USA. ✉email: stillman@cshl.edu

In the budding yeast *Saccharomyces cerevisiae*, replication origins are specified by DNA sequence motifs that comprise an essential A element (about 11nt in length) and multiple B elements[1]. Such sequences enable the replication of extra-chromosomal plasmids and are thus termed autonomously replicating sequences (ARSs)[2]. By contrast, replication origins are sequence non-specific in plants and animals, in the fission yeast *Schizosaccharomyces pombe*, and even in many other budding yeasts and fungi[3]. Nevertheless, the proteins involved in the initiation of DNA replication are highly conserved. In eukaryotes, the six subunit ORC complex (comprising Orc1–6) assembles on DNA prior to S-phase[4]. ORC then recruits Cell Division Cycle 6 (Cdc6), chromatin licensing and DNA replication factor 1 (Cdt1), and the replication helicase subunits Mcm2–7 to form a pre-replicative complex (pre-RC)[5]. In prior work, a structure of a pre-RC assembly intermediate containing the *S. cerevisiae* ORC-Cdc6-Cdt1-Mcm2–7 (OCCM) bound to origin DNA (*ARS1*) was determined at ~3.9 Å by cryo-electron microscopy[6]. This structure revealed multiple OCCM–DNA interactions, including an Orc4 α-helix inserted into the DNA major groove and an Orc2 loop inserted into the minor groove (Fig. 1a). These interactions were subsequently confirmed by a higher resolution ORC–DNA structure[7]. We note that a lysine-rich region of Orc1 interacts with DNA in this latter structure but not in the OCCM, suggesting considerable plasticity in origin recognition during pre-RC assembly.

Interestingly, the Orc4 α-helix and Orc2 loop have evolved in a manner that parallels the evolution of origin sequence specificity. Sequence alignments suggest that these features have been acquired in a sub-group of *Saccharomyces*-related budding yeasts, but are absent in all other eukaryotes including other budding yeasts, other fungi (including *S. pombe*), plants, and animals (Fig. 1c, Supplementary Fig. 1a). High-resolution structures of human[8] and *Drosophila*[9] ORC show the lack of the Orc4 α-helix and Orc2 loop (Fig. 1b, Supplementary Fig. 1b). The Orc4 α-helix and Orc2 loop are present but diverged in some other budding yeasts, such as *Kluyveromyces lactis*, which has sequence-specific origins that exhibit a DNA sequence motif that differs from the *S. cerevisiae* motif. These observations suggest that the Orc4 α-helix and/or Orc2 loop might play key roles in origin sequence specificity.

We demonstrate using two independent assays that the Orc4 α-helix and Orc2 loop are required for the initiation of DNA replication and that the Orc4 α-helix contributes to determining DNA sequence-specific origins in the yeast *S. cerevisiae*. By examining ORC sequences in a wide variety of eukaryotes for the presence of these origin specification domains, we find that the specification of DNA replication origins by the Orc4 α-helix and Orc2 loop co-evolved with ORC-Sir4-dependent transcriptional gene silencing and the loss of RNA interference (RNAi).

## Results

**Orc4 α-helix and Orc2 loop mutagenesis and characterization.** To investigate which specific residues might be involved, 32 individual Orc4 α-helix mutants and 7 Orc2 loop mutants were examined using a plasmid shuffle assay (Fig. 1d, e, Supplementary Figs. 2, 3; see "Methods" section). The Orc2 loop mutants were either lethal, had strong defects, or had little effect (Fig. 1e, Supplementary Fig. 3). Deletion of the Orc4 α-helix (from A481 to Q493) or its replacement with the 13-amino acid *K. lactis* Orc4-α-helix were lethal. As shown in Fig. 1 and noted below, *S. cerevisiae* and related species most likely acquired the α-helix during Orc4 evolution as part of the mechanism to locate origins at specific DNA sequences, and

hence loci within the genome, whereas species that do not have the α-helix must have other mechanisms to locate pre-RC assembly at sites along chromosomes. Other Orc4 α-helix mutants led to different levels of growth deficiency (Fig. 1d, Supplementary Fig. 2). Based on the growth deficiency phenotype, nine viable *ORC4* mutants were chosen for further detailed analysis. To perform the genetics in a complete manner, two conservative mutations ($orc4^{F485Y, \ Y486F}$ and $orc4^{R478K}$) were chosen for comparison. The wild type (WT) and *ORC4* mutants were tagged at the amino terminus (NTAP-tag) and the gene integrated into the genome as the sole locus to form a functional ORC (Supplementary Fig. 4). Some strains with the integrated version of the mutant *ORC4* (strains G, Supplementary Fig. 6) proliferated far better compared to strains that relied on a mutant ORC4 subunit that was expressed from a single origin minichromosome (strains P, Supplementary Fig. 6). Five of these mutants were created to investigate the FY residues at positions 485–486, which have evolved to IQ in *K. lactis*. Specifically, $orc4^{Y486Q}$, $orc4^{F485I}$, and $orc4^{F485I, \ Y486Q}$ were used to study the effects of these evolutionary changes, while $orc4^{F485A, \ Y486A}$ and $orc4^{F485Y, \ Y486F}$ (a conservative swap of aromatic amino acids) were used to investigate these residues more generally. The other four mutants, $orc4^{R478A}$, $orc4^{R478K}$, $orc4^{N489A}$, and $orc4^{N489W}$, were chosen to investigate the roles of R478 and N489, two conserved residues at opposite ends of the α-helix that we predicted to mediate both protein–protein contacts and contacts with DNA backbone phosphates. Some of these *ORC4* mutants exhibited slower growth rates (Fig. 1f) and slowed passage through S phase and mitosis (Supplementary Fig. 5). To better understand the effects of these mutations on origin activity and specificity, we performed two complementary, but independent deep-sequencing-based assays: massively parallel origin mutagenesis on plasmids and genome-wide DNA replication profiling.

**Massively parallel origin mutagenesis.** To quantify the sequence-dependent activity of ORC at specific origins of interest, we performed a massively parallel origin selection assay (MPOS assay) on two different origins in WT and nine yeast strains harboring the *ORC4* variants. 150 base-pairs of either *ARS1* (also known as *ARS416*) or *ARS317* DNA were synthesized at a 15% per-nucleotide substitution rate and cloned into plasmids that carried a selective marker[10,11] (Fig. 2a). These two plasmid libraries were then separately transfected into the 10 strains of yeast described above, and the mutated ARSs that remained after multiple cell divisions were sequenced. A custom motif inference algorithm, based on mutual information maximization (IM), was then applied to these sequence data and used to infer quantitative motifs describing origin activity in each strain. This algorithm proved to be essential for the correct analysis of the data (see below).

Some mutants, such as $orc4^{F485Y, \ Y486F}$ and $orc4^{R478K}$, yielded motifs very similar to WT (Fig. 2b for *ARS1*, Supplementary Fig. 7a for *ARS317*). Other strains, such as the $orc4^{N489A}$, $orc4^{N489W}$, $orc4^{R478A}$, and $orc4^{F485I, \ Y486Q}$ retained a far less diverse set of mutant ARSs and yielded noisier motifs that, nevertheless, remained relatively similar to the WT ARS consensus sequence (ACS) (Fig. 2b for *ARS1*, Supplementary Fig. 7a for *ARS317*). However, two Orc4 α-helix mutants exhibited robust changes to their ARS motifs: in both the $orc4^{F485A, \ Y486A}$ and $orc4^{Y486Q}$ mutants, and for both the *ARS1* and *ARS317* experiments (Fig. 2b and c, Supplementary Fig. 7), two dinucleotides present in the WT consensus sequence motif at positions 29–30 changed. In the *ARS1* experiments, motif A/T

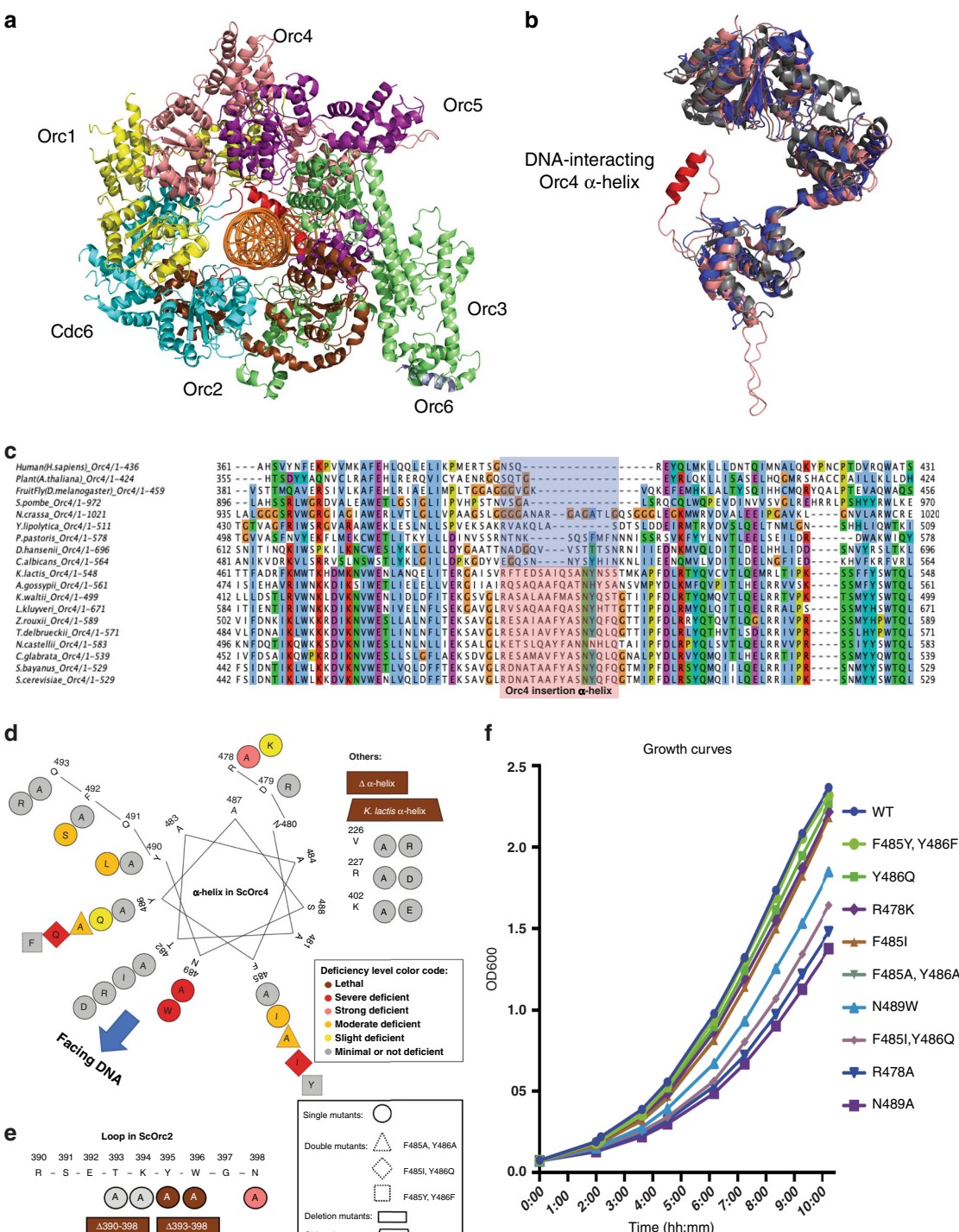

**Fig. 1 DNA interacting Orc4 α-helix and Orc2 loop are essential. a** Top-view of ORC–Cdc6 structure encircling an origin DNA with Orc1–6 and Cdc6 indicated. Recolored from previous cryo-EM work[6] OCCM structure (PDB code 5udb) and Orc4 α-helix and Orc2 loop that interact with DNA are colored in red. **b** Orc4 structure superposition of human Orc4 in blue (from PDB code 5uj7), *Drosophila* Orc4 in gray (from PDB code 4xgc) and *S. cerevisiae* Orc4 in salmon (from PDB code 5udb). Orc4 α-helix that interacts with DNA is colored in red. **c** Multiple sequence alignment of Orc4 among representing eukaryotic species as indicated. Orc4 α-helix region indicated with species that do not have sequence-specific origins shadowed in blue and species that sequence-specific origins exist shadowed in pink. **d, e** Maps of mutant viability phenotypes from the plasmid shuffle assay: Orc4 α-helix in a helical wheel **d** and Orc2 loop in connected line **e**. Mutant deficiency phenotypes (Supplementary Figs. 2 and 3) are summarized in color codes as indicated. Amino acids indicated in one-letter abbreviation. Different mutant types are indicated with different shapes. **f** Growth curves of NTAP-Orc4 integrated strains (see "Methods" section) in YPD with initiation OD600 at 0.05 at 30 °C to measure OD600 at different time points.

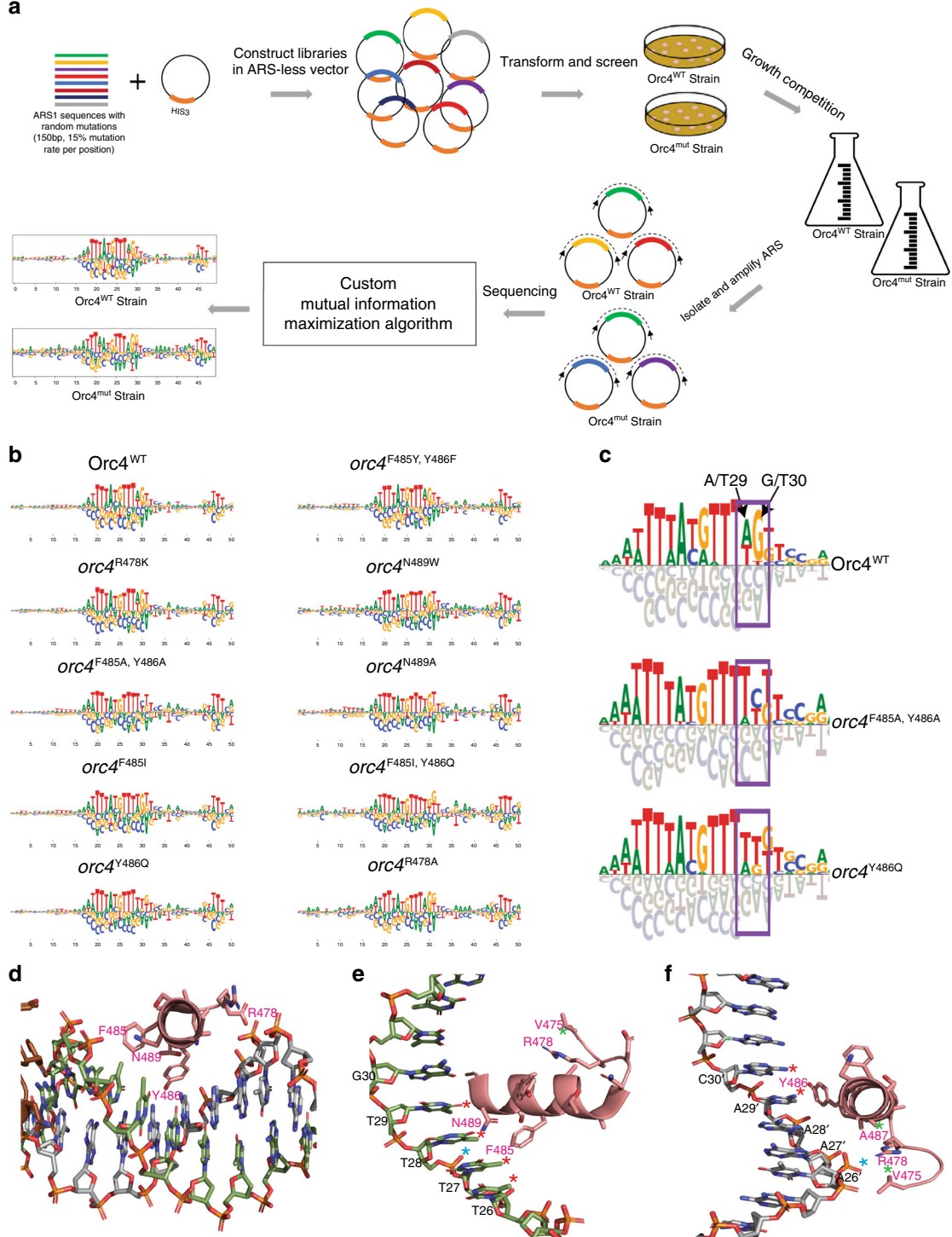

**Fig. 2 Selected origin sequence changes using a massively parallel origin selection (MPOS) assay. a** Schematic diagram of the MPOS assay. **b** ARS motifs for Orc4-integrated variants at A and B1 elements generated using an *ARS1 (ARS416)* variant library. See "Methods" for how motifs are graphically rendered. **c** Magnified view of the A element region in **b** from *Orc4*^WT, *orc4*^F485A, Y486A, *orc4*^Y486Q strains. Dark purple rectangles indicate the major changes at positions 29–30 in the Orc4 mutant strains. **d** Top-view of Orc4 α-helix insertion from ORC–DNA structure at 3 Å (PDB code 5zr1) positioned in the DNA major groove. F485, Y486, N489, and R478 interact with DNA in base-specific (specificity) and base-nonspecific (affinity) manner. **e** and **f** Same as in **d**, but viewed at different angles. Red asterisks denote the base-specific interactions between amino acid and DNA base. Blue asterisks denote the base-nonspecific interaction between amino acid and DNA phosphate backbone. Green asterisks denote the interaction between amino acids. Prime symbols denote bases on the opposite strand. Bases numbering denotes the positions in the logo (see **b**). **e** Shows the hydrophobic interaction between F485 and T-rich region T26–T29, base-specific interaction between N489 and T28, base-nonspecific interaction between N489 and phosphate backbone of T28, and R478 interaction with V475. **f** Shows the aromatic edge-face interaction between Y486 and A29′ on the opposite strand, hydrophobic interaction between Y486 and C30′, base-nonspecific interaction between R478 and phosphate backbone of A27′, and R478 interaction with A487 and V475.

G/T has been switched to T/A C/T in the $orc4^{F485A, Y486A}$ strain and switched to T/A T/G in the $orc4^{Y486Q}$ strain (Fig. 2c). Substantial changes were observed at the same position in the *ARS317* MPOS assay (Supplementary Fig. 7b). A principal component analysis (PCA) of the motifs inferred from multiple biological replicates confirmed that reproducible changes in motifs indeed resulted from the mutations in question (Supplementary Fig. 8a). A quantitative analysis of motifs inferred using IM, realtive to those inferred using standard enrichment ratio (ER) calculations, showed that the increased sensitivity afforded by IM was essential for resolving these mutation-dependent changes in origin specificity (Supplementary Fig. 8).

**The structural basis for origin sequence specification**. Observations using the ~3 Å high-resolution structure of Orc4 α-helix on DNA (Fig. 2d, Supplementary Movie 1), from the structure of ORC on DNA[7], can further rationalize the mutation-dependent change in origin specificity. We suggest that Y486 interacts with the DNA base C/A30 (Fig. 2f), which is on the complementary strand from the A/T G/T dinucleotide whose readout is altered in the $orc4^{F485A, Y486A}$ and $orc4^{Y486Q}$ mutants (Fig. 2c, purple box) in a face-to-edge T-type π interaction, as often seen in protein–DNA interfaces[12]. In addition, F485 sits against a hydrophobic stretch comprised of the methyl groups emanation from a run of T's, T26–29 (Fig. 2e).

At each end of the Orc4 insertion α-helix are amino acids that we suggest provide affinity for ORC to DNA by binding to the phosphate backbone as well as help position this sequence-reading α-helix correctly in the major groove. R478 interacts with A487 on the α-helix as well as the adjacent V475 and could have an alternative conformation whereby it contacts a DNA phosphate (Fig. 2e, f). Even a conservative amino acid substitution R478K was slightly deficient (Fig. 1f and Supplementary Table 1) and had a subtle change in the genome wide origin firing pattern (Supplementary Fig. 11d and Fig. 3c) indicating that there is an additional role for the R478 side chain, and that the charge and length of the lysine side chain cannot fully replace the arginine at this location. Likewise, N489 that sits at the opposite end of the α-helix would be in range of a DNA phosphate contact and even a DNA base contact with T28 upon minor adjustments of the model that are well within the EM density[7] (Fig. 2e). Mutation of either amino acid had the largest, non-lethal effect (Fig. 1) and thus we suggest that they could play key roles in both positioning the α-helix in the major groove and contributing to affinity of ORC to DNA.

**Genome-wide replication origin profiling**. To investigate the mutation-dependent origin usage changes in natural genomic replication origin profiling, cells were arrested in G1 phase and released into S phase in the presence of hydroxyurea (HU). HU treatment restricts (via checkpoint signaling) origin firing to those origins that normally become active in early S phase, and prevents the activation of origins that normally fire later. The addition of 5-Ethynyl-2′-deoxyuridine (EdU), followed by purification of EdU-labeled DNA and high-throughput DNA sequencing, was then used to map the locations and activities of early origins throughout the genome (Supplementary Fig. 9 and Fig. 3c). Mrc1 is a replication fork associated protein and mediator of intra-S phase checkpoint signaling. All origins fired, as expected, when the *MRC1* gene was deleted. We observed more origins firing than have been confirmed in oriDB[13]. This $ORC4^{WT}$, $mrc1\Delta$ profile thus reveals a maximal set of possible origins against which to compare the replication profiles of NTAP-tagged Orc4-integrated strains. As expected, only a subset of the maximal set of possible origins fired in WT cells containing the NTAP-tagged $Orc4^{WT}$ protein. The genome-wide replication origin profiles were reproducible in biological replicates (Supplementary Fig. 9b for WT and $mrc1\Delta$; Fig. 3c and Supplementary Fig. 9c for all mutant Orc4 strains).

Only two completely de novo replication origin locations were found in genomic origin firing profiles of all nine orc4 mutant strains, which were neither predicted to be ARS locations in OriDB nor exist in the $ORC4^{WT}$, $mrc1\Delta$ profile. One was found in $orc4^{F485I, Y486Q}$ (Fig. 3a) and another was found in $orc4^{F485A, Y486A}$ strain (Fig. 3b). Indeed, we did not expect to see a dramatic change in de novo origin locations because the mutations we made are only single or double point-mutations and would not be expected to create many new replication origin locations. However, extensive changes in the Orc4 α-helix, such as $orc4^{Klα-helix}$ or $orc4^{\Delta\alpha-helix}$, were lethal and therefore could not be assessed for de novo origins.

Despite few de novo replication origin locations, the genomic origin firing pattern changed considerably in some strains (Supplementary Fig. 11b–j). There were numerous origins that were active in both the $mrc1\Delta$ and the wild-type strain (aka. early origins) but were specifically repressed in the orc4 mutant strains (Fig. 3f and Supplementary Fig. 11a–j, orange arrow direction). At the same time, there were numerous origins that were inactive in the wild-type strain but were activated in the orc4 mutant strains (Fig. 3c, e, h, green arrows and Supplementary Fig. 11a–j, green arrow direction). The activation or repression pattern is mutant dependent. The chromosome IV profiles for the nine orc4 mutant strains was used as an example chromosome to show more detail (Fig. 3c, Supplementary Fig. 9c). While some origins, either those that normally fire early or late, did not change (Fig. 3c, g, i, black arrows), other origins changed their firing pattern in orc4 mutant strains. Examples are indicated by green arrows (Fig. 3f, e, h, as explained above). Interestingly, two origins on chromosome IV were active only in the $orc4^{F485I, Y486Q}$ mutant strain, in which the IQ are the amino acids that exist in *K. lactis* (Fig. 3c, e, i, red arrows). When, however, the conservative $orc4^{F485Y, Y486F}$ double mutant was analyzed, the firing pattern was like WT.

There are chromatin and chromosome location context factors that were suggested in previous studies to play roles in controlling the origin timing[14,15]. Our quantitative genome-wide analysis supports this idea. Perhaps surprisingly, origin peak heights generally do not correlate with how well the origin sequence matches the ACS motif (Supplementary Fig. 12, section A). Only when origin sequence recognition is reduced/disturbed, such as in the orc4 F485 and Y486 mutants, are origin locations that have high peaks in the wild type strain affected (Supplementary Fig. 12, section B). The orc4 R478 and N489 mutations had a slower growth phenotype and this is likely due to reduced origin licensing, consistent with the proposal that the Orc4 α-helix that is not positioned correctly due to amino acid changes at each end of the α-helix, making it more difficult to stably position within the DNA major groove. As a consequence, poor DNA interaction would lead to overall much fewer origins that become active (Supplementary Fig. 11q–t support this idea) resulting in larger replicon size (i.e., greater inter-origin distance) and a slower doubling time (Fig. 1f). These mutants would be forced to use late origins under HU to survive. This is also consistent with the observation that the peak widths on average for these slow growing mutants was larger than WT peak heights because the smaller number of origins utilized would not be restrained by rate-limiting replication factors, which are known to exist[16,17]. We suggest that the mutations at Orc4 R478 and N489 would

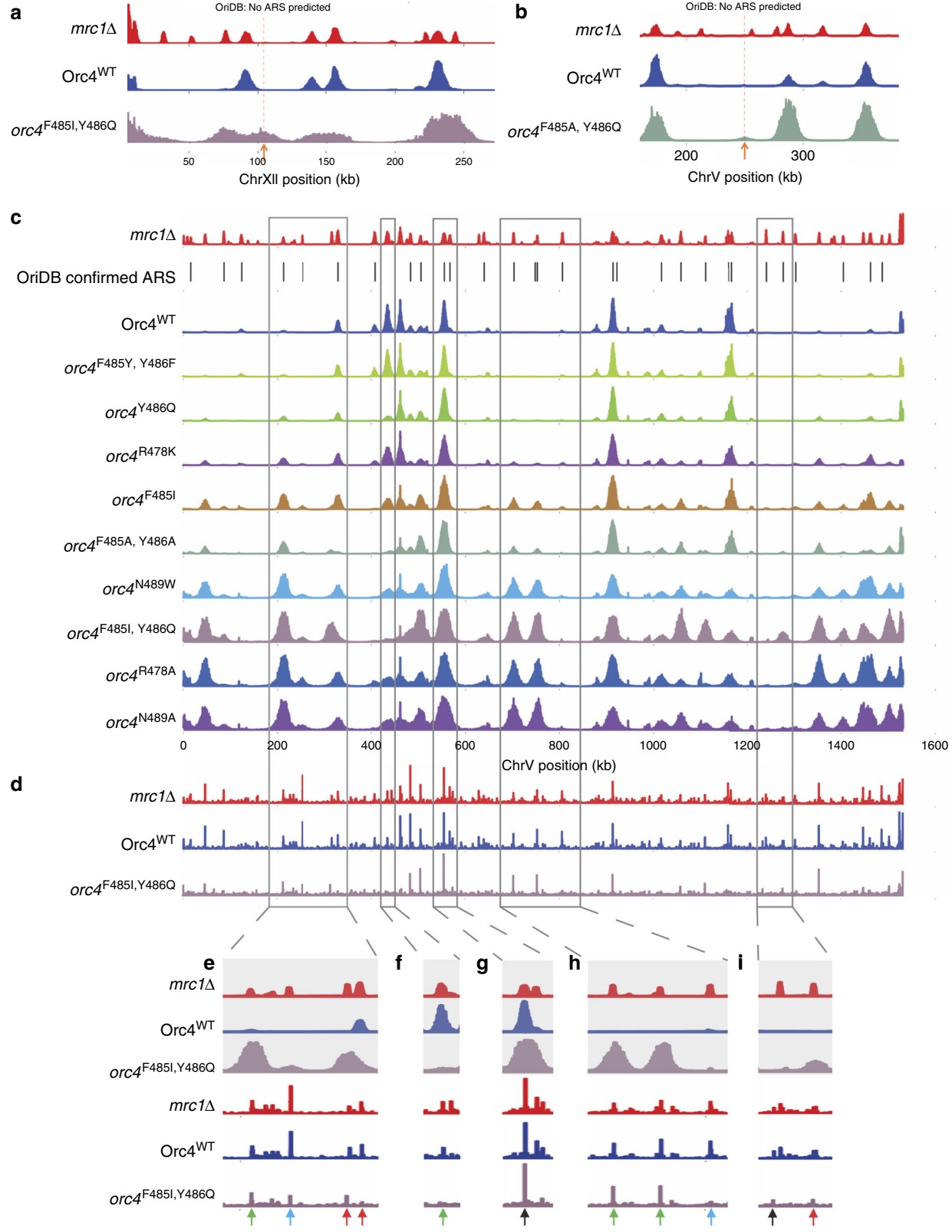

not cause sequence specificity changes but instead cause inefficient binding to origins. Indeed, the correlation between peak heights from genome-wide origin data and matching scores to motifs derived from the MPOS *ARS* selection assays did not correlate well in the *orc4* R478 and N489 mutant strains compared to strains (e.g., *orc4*[F484I, Y486Q] and *orc4*[F485A, F486A]) that change DNA sequence specificity (Supplementary Fig. 12,

section C). In addition to the known influence of chromatin context on origin timing[14,15], one possible explanation for these results is that the dramatic changes in origin utilization in the genome in the *orc4* mutants are a result of altered affinity and sequence specificity of ORC and ORC—Cdc6 for origin DNA.

The recruitment of Mcm2–7 to DNA replication origins was analyzed by Mcm2 chromatin-immunoprecipitation (ChIP) in

**Fig. 3 Orc4 α-helix mutants change the pattern of origin firing and MCM binding.** Genome-wide origin firing profile from yeast that were α-factor blocked and released into S phase in 200 mM hydroxyurea (HU) for 90 mins. **a** and **b** Orange arrows and dotted lines indicate the de novo origin locations. **a** shows a new origin at location ChrXII: 95,827-117,318 in the orc4[F485I, Y486Q] strain. **b** shows the completely new origin location ChrV: 248,069–251,910 in orc4[F485A, Y486A] strain. **c** Whole genome replication profiles. Chromosome IV (ChrIV) is shown as a representation. Strains are in the order of shorter to longer doubling time (Supplementary Table 3) from top to bottom. **d–i** ChIP profile of MCM (anti-Mcm2) of NTAP-Orc4 integrated strains at ChrIV in G1 phase and it's comparison to replication origins profile. orc4[F485I, Y486Q] strain is used as an example of NTAP-Orc4 mutant strains to compare with Orc4[WT] strain in whole-chromosome view **d** and zoomed-in views **e–i**. Green arrows indicate the example locations of origins with firing pattern changes in the orc4[F485I, Y486Q] strain. At these locations, similar changes were also observed in other mutants (see **c**). Black arrows indicate the example locations of origins with firing patterns that remained the same in orc4[F485I, Y486Q] strain. At these locations, origins with firing pattern also remained the same in other mutants (see **c**). Red arrows indicate the example locations of origin firing pattern changes that are unique in the orc4[F485I, Y486Q] strain but not in other mutant strains (see **c**). Blue arrows indicate the examples of inactive late origins in HU in WT that have lower or no Mcm2-ChIP signal in the orc4[F485I, Y486Q] mutant strain and are not active origins in the mutant strain.

G1 phase (Fig. 3d, Supplementary Fig. 10b). The Mcm2-ChIP results were reproducible in biological replicates (Supplementary Fig. 10a) and correspond well to the location of active and potentially active DNA replication origins, although we note that there are other Mcm2 ChIP peaks not associated with origins (Fig. 3d). For example, it is known that Mcm2–7 can move on chromosomes once loaded[18]. Nevertheless, in the mrc1Δ strain where all potential DNA replication origins were active, the Mcm2-ChIP profile corresponded well with the DNA replication profile (Fig. 3a, d). As expected, in the presence of HU, late origins did not fire in the Orc4[WT] strain, even though there was Mcm2–7 bound. Indeed, the Orc4[WT] Mcm2-ChIP profile was similar to the profile in the mrc1Δ strain (Fig. 3d). We then compared the replication and Mcm2-ChIP profiles in the orc4[F485I, Y486Q] mutant strain to the profiles in the Orc4[WT] and mrc1Δ strains. While in general the Mcm2-ChIP profile in the orc4[F485I, Y486Q] mutant was similar to the profile in the mrc1Δ strain, there were changes were observed in the replication profile, the Mcm2-ChIP profile, or both. For example, some Mcm2-ChIP peaks that did not change were present at late origins that fired in the orc4[F485I, Y486Q] and mrc1Δ mutant stains but not in WT (e.g., Fig. 3e, h, green arrows). Some of the orc4 mutant Mcm2-ChIP peaks did change along with the replication profile when compared to WT and the mrc1Δ mutant. For example, both the active origins and the ChIP-Mcm2 peaks were absent in the orc4[F485I, Y486Q] mutant strain, but were present in the WT and the mrc1Δ mutant strains (Fig. 3f, green arrow). In some cases, the Mcm2-ChIP and replication profiles did not change in the orc4[F485I, Y486Q] mutant compared to WT (e.g., Fig. 3g, i, black arrows). In other cases, the orc4 mutant lost Mcm2-ChIP peaks (e.g., Fig. 3h, blue arrow). A particularly interesting case is the origin switch in the orc4[F485I, Y486Q] mutant strain (Fig. 3d, e, two red arrows at right), where the Mcm2 binding also switched, albeit not completely. Combined, these results suggest that the Orc4 α-helix is a major determinant of origin utilization in the genome.

**DNA sequence statistical analysis for genome-wide origin firing patterns.** We examined the DNA sequences of the ACSs from OriDB[13,19] under each EdU peak and performed a genome-wide statistical analysis. The result shows that, specifically in the Y486 mutant strains (orc4[F485I, Y486Q], orc4[F485A, Y486A], and orc4[Y486Q]), the origin firing peak heights were significantly reduced when the dinucleotide "AG" occurred at nucleotide position 29–30 of the aligned MPOS-derived motifs (Figs. 2c and 4a, c–e). Hereafter, these positions in the genomic ACS are referred to as "position 29–30" for easier reference. The Orc4[WT] strain, together with the rest of the mutant strains, have relatively equal origin firing peak height regardless of the origin dinucleotide sequence at positions 29–30 (Fig. 4a, b, Supplementary Fig. 13). Thus, both the MPOS

assay data (Fig. 2c) and the analysis of the ACS in the origins used in the genome, show that Orc4 F485 and Y486, especially Y486, are essential for recognizing origins with the "AG" dinucleotide sequence at positions 29–30. These mutants effectively reduce the chances of utilizing origins with the "AG" sequence. These data show that the Orc4 α-helix defines the sequence specificity and hence location of active origins in the genome.

**Co-evolution of origin specification and gene silencing mechanisms.** The data using both whole genome analysis and the MPOS assays demonstrate that the Orc4 α-helix contribute to selection of origin sequences in the yeast genome, as predicted by the structure of the OCCM[6] and ORC[7] on origin DNA. The conservation of the α-helix and loop is restricted to a small clade of Saccharomyces-related species and where it has been determined, corresponds to the origins of DNA replication having a demonstrable consensus sequence (Raguraman, M.K. and Liachko, I. in Kaplan[20]) (Fig. 5).

It is known that in some budding yeasts, such as S. cerevisiae and K. lactis, ORC, functioning with silent information regulator (SIR) proteins, is also required in transcriptional gene silencing of mating type loci, rDNA and telomeres[21–23]. Evolutionarily, Sir2 and Sir4 preceded the acquisition of Sir1 and Sir3 (which is related to Orc1) in the ORC-Sir4-mediated transcriptional gene silencing pathway. In K. lactis, Sir4 binds directly to Orc1 but in S. cerevisiae, Sir1 binds to Orc1 and Sir4 binds to Sir3, which is encoded by SIR3 that arose from the ancestoral ORC1 gene as a result of whole genome duplication[23] (WGD, Fig. 5). In both species Sir2 is required because its histone deacetylase activity is essential for the gene silencing. Interestingly, Sir4 binds to Esc1 which is located in the nuclear envelope, tethering the silent loci to the nuclear periphery[24]. Of relevance here is that Sir4 is related in structure to nuclear lamins, which are present in most eukaryotes but are absent in yeast[25].

We observed a very interesting co-evolution of origin sequence specification (Fig. 5, first three columns) and gene silencing (Fig. 5, remaining columns). The acquisition of sequence-specific origins, the Orc4 α-helix and the Orc2 loop (Fig. 5, blue shadow) correlated precisely with the acquisition of ORC-Sir4-mediated transcriptional gene silencing (Fig. 5, yellow shadow).

Dicer is an RNase III family member and a key mediator in the RNAi pathway, which has been shown to control gene silencing by transcriptional and post-transcriptional mechanisms[26]. However, Dicer, but not other components of the RNAi pathway, has an RNAi-independent role in S. pombe in the termination of transcription at replication stress sites[27]. This may contribute to alleviation of R-loop-mediated conflicts between DNA replication and transcription, particularly in repeated sequences and heterochromatin. The vast majority of eukaryotes that lack sequence-specific origins, including plants, animals, and the majority of fungi including yeast have vast repeated sequences

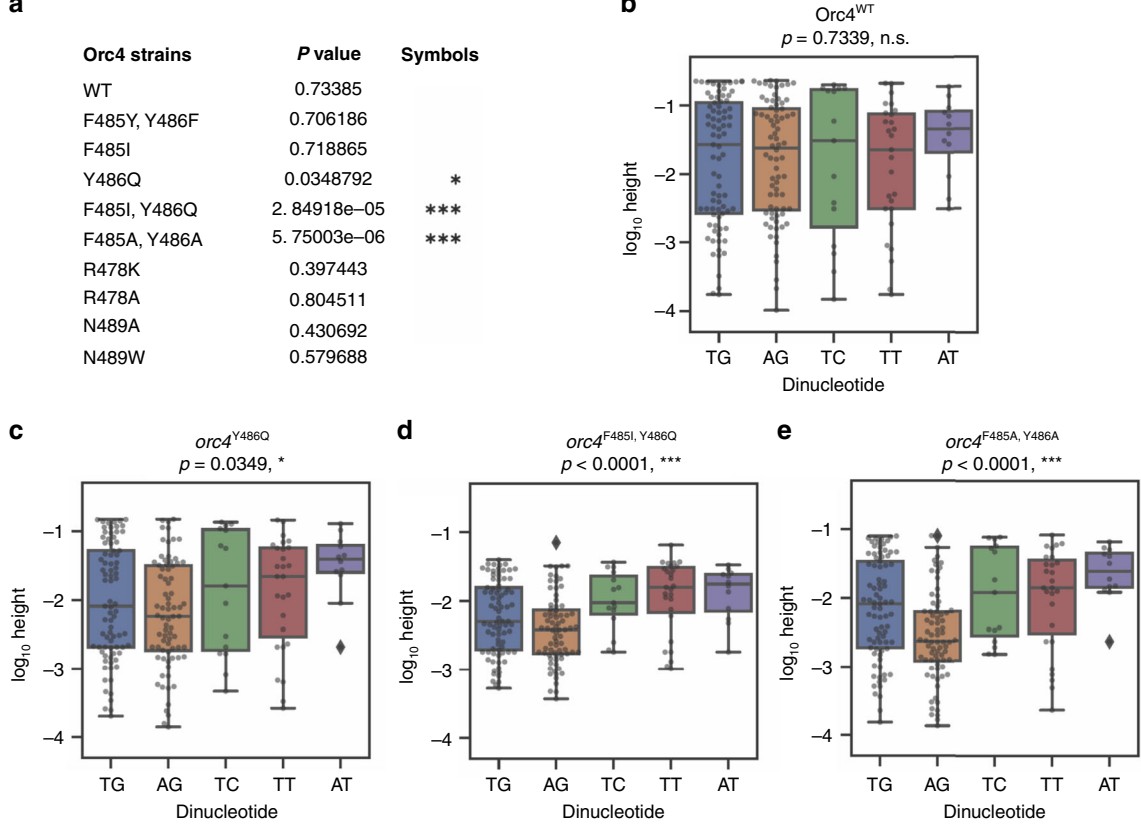

**Fig. 4 Genomic origin firing pattern changes are sequence specific.** DNA sequences under each of the origin replication peak that match the ARS consensus sequence (ACS) motif were obtained from OriDB[13,19]. Genome-wide statistical analysis was performed to check the dependence of origin firing peak height on dinucleotide identity at positions 29–30, which were chosen based on the results of our MPOS analysis (Fig. 2c), using a one-way ANOVA test for dependence. There were 5 dinucleotide variants that occurred 3 or less times in 9 annotated ACSs were removed prior to this analysis. In the remaining $n = 211$ annotated ACSs, there were 5 dinucleotide variants at 29–30; these were analyzed and the peak heights for each dinucleotide are plotted. Asterisks denotation: *$p < 0.05$, **$p < 0.01$, ***$p < 0.001$. **a** Genome-wide statistical analysis results for all ten strains. **b–e** Box plots for Orc4 wild-type (**b**) and three of the *orc4* mutant strains (**c–e**). Y-axis shows genomic $\log_{10}$ peak heights. Each dot denotes an annotated ACS. Box plots elements: the minimum height, first (lower) quartile, median, third (upper) quartile, and maximum height. Diamonds denotes outliers that exhibited aberrantly large values. The "AG" dinucleotide at position 29–30 is not utilized in mutants that change Y486 to alanine or glutamine.

and heterochromatin and thus need RNAi[26] (Fig. 5, orange shadow) or Dicer's RNAi-independent role in maintaining genome stability, particularly if origin locations are stochastic, as has been shown in *S. pombe*[28].

Budding yeasts that lack sequence-specific origins, such as the pathogenic yeast *Candida albicans* and the industrial yeast *Yarrowia lipolytica* that can metabolize unusual hydrocarbons, have lost, or are in the process of losing RNAi[29,30]. Some retain a non-canonical Dicer (Dcr*) that has an RNase III domain and has been shown in *C. albicans* to exhibit RNAi to silence transposable elements and sub-telomeric repeated sequences[29]. They lack both the Orc4 α-helix and the Orc2 loop and do not have demonstrable sequence-specific origins. In this context, *Y. lipolytica* is an interesting case since it has lost Dicer and Argonaute (Ago) and lacks ORC-Sir4 silencing and sequence-specific origins. *Y. lipolytica* has dispersed rDNA gene clusters that are sub-telomeric and has a relatively low gene density (one gene per 3.3 kb), far lower than the gene density found in *S. cerevisiae* (one gene per 2 kb)[31]. It also uses Tay1, a TRF-like protein for telomeric and sub-telomeric gene silencing, which is more similar to the human shelterin complex mechanism[32]. Moreover, it is a heterothallic yeast, which does not switch its mating type and thus lacks silent mating type loci[31]. We suggest and that *Y. lipolytica* may be a useful species to investigate origin

location and sequence specificity and we are studying replication patterning in this species.

The budding yeasts that have acquired sequence-specific origins and ORC-Sir4-mediated gene silencing system are likely to have lost RNAi completely, albeit some retained the non-canonical Dicer (Dcr*). One budding yeast, *T. delbrueckii*, has ORC-Sir4 silencing and has retained Dcr* and Ago, but the latter are not involved in transcriptional gene silencing[30]. As species lost RNAi with a concomitant reduction in repeated sequences in the genome, including centromere-associated repeated sequences, we suggest that in the *Saccharomyces*-related, ORC-Sir4-containing budding yeast that ORC evolved to bind DNA in a sequence-specific manner, providing a mechanism to locate origins of DNA replication in intergenic regions[5]. Such a location would help maintain genome stability by reducing the possibility of conflicts between DNA replication and transcription, including the formation of R-loops[33]. The remaining repeated sequences in these species, such as the silent mating type loci in homothallic yeast, have evolved to be protected from loss by recombination and be transcriptionally silenced by an ORC-Sir4-dependent recruitment of the histone deacetylase Sir2. It is possible that in *Y. lipolytica*, Sir2 binds directly to ORC and thus bypasses the requirement for the other SIR proteins.

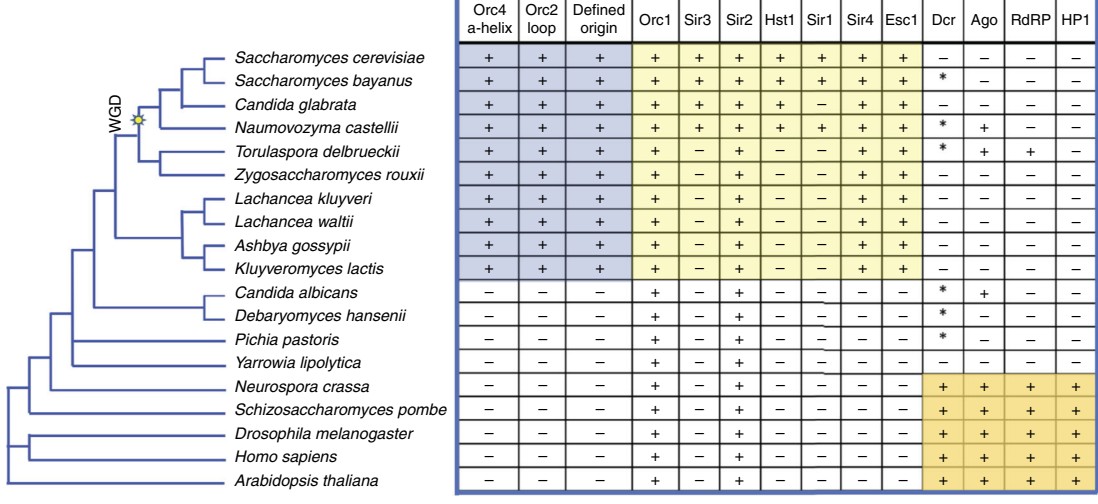

| | Orc4 a-helix | Orc2 loop | Defined origin | Orc1 | Sir3 | Sir2 | Hst1 | Sir1 | Sir4 | Esc1 | Dcr | Ago | RdRP | HP1 |
|---|---|---|---|---|---|---|---|---|---|---|---|---|---|---|
| *Saccharomyces cerevisiae* | + | + | + | + | + | + | + | + | + | + | – | – | – | – |
| *Saccharomyces bayanus* | + | + | + | + | + | + | + | + | + | + | * | – | – | – |
| *Candida glabrata* | + | + | + | + | + | + | – | + | + | + | – | – | – | – |
| *Naumovozyma castellii* | + | + | + | + | + | + | + | + | + | + | * | + | – | – |
| *Torulaspora delbrueckii* | + | + | + | + | – | + | – | – | + | + | * | + | + | – |
| *Zygosaccharomyces rouxii* | + | + | + | + | – | + | – | – | + | + | – | – | – | – |
| *Lachancea kluyveri* | + | + | + | + | – | + | – | – | + | + | – | – | – | – |
| *Lachancea waltii* | + | + | + | + | – | + | – | – | + | + | – | – | – | – |
| *Ashbya gossypii* | + | + | + | + | – | + | – | – | + | + | – | – | – | – |
| *Kluyveromyces lactis* | + | + | + | + | – | + | – | – | + | + | – | – | – | – |
| *Candida albicans* | – | – | – | + | – | + | – | – | – | – | * | + | – | – |
| *Debaryomyces hansenii* | – | – | – | + | – | + | – | – | – | – | * | – | – | – |
| *Pichia pastoris* | – | – | – | + | – | + | – | – | – | – | * | – | – | – |
| *Yarrowia lipolytica* | – | – | – | + | – | + | – | – | – | – | – | – | – | – |
| *Neurospora crassa* | – | – | – | + | – | + | – | – | – | – | + | + | + | + |
| *Schizosaccharomyces pombe* | – | – | – | + | – | + | – | – | – | – | + | + | + | + |
| *Drosophila melanogaster* | – | – | – | + | – | + | – | – | – | – | + | + | + | + |
| *Homo sapiens* | – | – | – | + | – | + | – | – | – | – | + | + | + | + |
| *Arabidopsis thaliana* | – | – | – | + | – | + | – | – | – | – | + | + | + | + |

Tree in unscaled branches

\* Non-canonical Dicer that cotains an RNase III domain (like RNT1)

**Fig. 5 Co-evolution of DNA replication origin specification and gene silencing.** Phylogenetic tree is drawn based on whole genome vicinity and is in unscaled branches. The whole genome duplication (WGD) event is indicated in the tree. Table inspired by previous studies on gene silencing[30]. Positive symbol indicates the exist of genes, negative symbol indicates the absent of genes, asterisk symbol indicates exist of noncanonical dicer. Orange box: species that use RNAi for gene silencing. Yellow box: species that use ORC and Sir proteins for gene silencing. Blue box: species that have sequence specific origins.

In *S. pombe*, Orc4 has an AT-hook DNA-binding domain at its amino-terminus that localizes initiation of DNA replication to AT-rich sequences in the genome[34] even though replication origin utilization throughout the genome is known to be stochastic[28]. We found similar sequences are present in Orc4 in many fungi, including *Neurospora crassa*. Since the AT-hook sequences and the Orc4 α-helix and Orc2 loop are absent in other fungi, animals and plants, they must have an alternative mechanism of specifying origin location, a topic of major interest.

ORC is involved in maintenance of heterochromatin in *Drosophila* and human, via an interaction between ORC1 and the heterochromatin protein HP1[35,36]. Furthermore, ORC in human cells is also involved in repression of transcription of the *CCNE1* gene encoding Cyclin E via interactions with the histone methyltransferase SUV39H1 and the Retinoblastoma tumor suppressor protein (Rb)[37]. Thus ORC-dependent gene silencing may exist outside of species that have acquired Sir4.

## Methods

**Yeast genetic methods and strain construction**. Yeast strains generated in this study (described in Supplementary Methods Table 1) were derived from W303-1a (*MATa ade2-1 can1-100 his3-11,15 leu2-3,112 trp1-1 ura3-1*). Primers and oligos sequences are described in Supplementary Methods Table 2.

The YB51 (*orc4Δ::TRP1+pORC4/URA3*) strain was used for plasmid shuffle assay (see "Methods", section "Plasmid shuffle assay"). A PCR-based gene deletion strategy was used for disrupting endogenous *ORC4* with *TRP1* and a *URA3*-containing plasmid (pRS416) carrying wildtype *ORC4* gene is used as complement.

The *ORC4* site-directed mutation constructs-containing plasmids were used for plasmid shuffle assay (see "Methods", section "Plasmid shuffle assay"). Based on a CEN-based *LEU2*-containing plasmid constructs (pRS415) carrying wildtype *ORC4* gene, *ORC4* site-directed mutation constructs were created using PCR mutagenesis strategy, confirmed by DNA sequencing (see "Methods", section "Plasmid shuffle assay").

The NTAP-Orc4-integrated yeast strains were used for phenotype characterization assays, including the genome-wide DNA replication origin profile analyses, ChIP and massively parallel origin mutagenesis and selection assay. NTAP-Orc4 integrated strains were derived from YB1588 (*MATa orc4Δ::TRP1 bar1Δ::TRP1 LEU2::BrdU-Inc+pORC4/URA3*), which is a meiotic product of a diploid strain obtained by crossing YB51 (*MATα orc4Δ::TRP1+pORC4/URA3*) and YB1549 (*MATa bar1Δ::TRP1 LEU2::BrdU-Inc*). YB1549 was derived from YS2251

(*MATa bar1Δ::TRP1*) by inserting a BrdU-INC cassette[38] with *LEU2* to facilitate EdU incorporation.

NTAP-Orc4 construct was used for NTAP-Orc4-integrated strains construction, which was generated using a PCR-based strategy with tag coming from pBS1761[39] (purchased from Euroscarf). The construct is inserted into *his3* locus of YB1588 using CRISPR/Cas9 system[40]. Then, the plasmid containing Cas9 gene was dropped off by non-selective culture and tested for loss of plasmid marker. Subsequently, the plasmid carrying wildtype Orc4 gene was dropped off by counter selecting on 5 fluoroorotic acid 5-FOA plates for loss of *URA3*. The loss of pORC4/*URA3* were confirmed by PCR and sequencing in combination with phenotypic assessment.

**Plasmid shuffle assay**. The Orc4 α-helix mutants were screened for function in vivo using plasmid shuffle assay[41]. The Orc4 site-directed mutation constructs-containing plasmids with LEU2 marker were transformed into YB51 (*orc4Δ::TRP1 +pORC4/URA3*) and selected on SC-Leu-Ura plates. The transformants were isolated, grown in YPD overnight, and spotted onto 5-FOA plates with 10-fold serial dilutions starting from $1.5 \times 10^7$ cells to select for loss of *URA3* plasmid carrying the wild-type Orc4. As control, the same dilutions were spotted on YPD plates. YPD or 5-FOA plates were cultured under 30, or 25 or 37 °C to test their cold or temperature sensitivity.

**Cell extract preparation, immunoprecipitation, immunoblot analysis, and antibodies**. Whole cell extracts from NTAP-Orc4-integrated strains (see "Methods", section "Yeast genetic methods and strain construction") was prepared[41]. Cell extracts were analyzed for protein concentrations. Immunoprecipitation procedures were performed by mixing ~1.6 mg of total proteins and 30 μl of the IgG Sepharose 6 Fast Flow beads (GE Healthcare, Cat# 17-0969-01) at 4 °C for 2 h and precipitating the NTAP-tagged Orc4. The beads were washed extensively with EBX buffer (recipe same as previously published[41]) and boiled in 30 μl loading sample buffer (1.0% β-Mercaptoethanol, 0.2% bromophenol blue, 20% glycerol, 100 mM Tris–Cl (pH 6.8), 4% sodium dodecyl sulfate). Proteins from immuno-precipitation (IP) and cell extract (as IP input) were fractionated by SDS–10% PAGE and transferred to nitrocellulose membrane. Immunoblot analysis was performed using antibodies against Orc4 (SB12) used at 1:2000 dilution and Orc1 (SB13) used at 1:1000 dilution and TBS with 0.05% Tween 20 was used for pre-paring blocking and washing solutions.

**Cell growth, block, synchronization, and flow cytometry analysis**. Exponentially growing yeast cells (~$10^7$ cells/ml) in YPD were synchronized in G1 with 25 ng/ml of α-factor (*bar1Δ* strains are used in this study) for 3 h at 30 °C. To release from G1 arrest, cells were collected by filtration and promptly washed twice on the filter using one culture volume of $H_2O$ and then resuspended into YPD medium. 1 ml of cells was collected at different time points by adding sodium azide to final concentration at 0.1%. Cells are quickly centrifuged, resuspended with 400

μl $H_2O$ and fixed by adding 1 ml 100% ethanol and rotate overnight at 4 °C. Cells then are quickly centrifuged, washed one time with $H_2O$, resuspended in 250 μl RNaseA (Sigma-Aldrich) solution (2 mg ml$^{-1}$), incubated for 4 h in a 37 °C shaker and then sonicated using a Tekmar Sonic Disruptor with 630-0418 Tapered Microtip for two cycles of pulse for 1 s "ON", 1 s "OFF" at amplitude setting 22–25%. Proteinase K (Sigma-Aldrich) solution was added to final concentration at 1 mg ml$^{-1}$ and incubate for 1 h in 50 °C in Eppendorf thermomixer R mixer, 1.5 ml Block with speed at 750 rpm. Cells were then quickly centrifuged, resuspend in 50 mM Tris pH 7.5 with SYBR green I (Thermo Fisher) diluted at 1:10,000 ratio and filtered through strainer cap tubes (Corning™ Falcon™ Test Tube with Cell Strainer Snap Cap). BD LSRFortessa Dual Special Order System instrument and BD FACSDiva Software Version 8.0.1 Firmware Version 1.4 (BD LSRFortessa) were used to collect the data by measuring SYBR green signal. Same number of yeast cells data (30,000 events per run) were collected for each sample. FlowJo Version 10.6.1 was used to analyze the data and no gating strategy used.

**MPOS assay**. Both the *ARS1* (*ARS416*) and HMR-E (*ARS317*) libraries used ARS sequences 150 bp in length and synthesized with a 15% mutation rate at each position. Variant ARSs were cloned in bulk into a *HIS3*-containing plasmid. The libraries were then transformed into NTAP-Orc4-integrated yeast strains (see "Methods", section "Yeast genetic methods and strain construction"). The transformed cells were plated on SC-his plate, grown in a 30 °C incubator, washed off from plates when saturated, inoculated into SC-his medium, shaken at 30 °C shaker until saturation and harvested. ARS-containing plasmid DNA was isolated, PCR-amplified, ligated with custom inline barcodes (Supplementary Methods Table 2), quantified, pooled, and submitted for sequencing. Computational analyses of MPOS assay data are described below (see "Methods", section "Computational analyses of MPOS data", and section "Code availability"). DNA-sequencing data have been deposited on the Sequence Read Archive database (see "Data availability").

**Computational analyses of MPOS data**. Illumina reads from the MPOS experiments were analyzed using custom Python scripts. The output of this pipeline was, for each library or selected sample, a list of variant ARS sequences with each sequence assigned a corresponding read count. These lists were used as input to both the ER and IM motif modeling algorithms described below.

Our ARS motif modeling effort aimed to predict the activity of a variant ARS based on its DNA sequence. Specifically, we sought a mathematical function $a(s)$ that quantifies the activity of an arbitrary input ARS DNA sequence $s$. We assumed this function could be represented by a matrix model[42], i.e.,

$$a(s) = \sum_b \sum_l \theta_{bl} s_{bl} \qquad (1)$$

where $b = A, C, G, T$ indexes the four DNA bases, $l = 1, 2,\ldots, L$ indexes nucleotide positions, the sequence $s = \{s_{bl}\}$ is represented by a $4 \times L$ matrix of indicator variables ($s_{bl} = 1$ if base $b$ occurs at position $l$; $s_{bl} = 0$ otherwise), and $\theta = \{\theta_{bl}\}$ is a $4 \times L$ matrix of model parameters that must be inferred from data. All inferred motifs were limited to sequences of length $L = 50$ encompasing both the A and B1 elements of the assayed ARSs. To facilitate the comparison of motifs to one another, both visually and through PCA analysis, motif parameters were centered and rescaled via the transformation $\theta_{bl} \rightarrow \theta'_{bl}/C$ where $\theta'_{bl} = \theta_{bl} - \frac{1}{4}\sum_{b'} \theta_{b'l}$ and $C = \sqrt{\sum_b \sum_l \theta'^2_{bl}}$.

Sequence logos were generated by Logomaker[43]. In these representations of motif parameters, the value of $\theta_{bl}$ is represented by the height of character $b$ at position $l$ (or negative that height if the character is drawn below the $x$-axis).

The PCA shown in Supplementary Fig. 8 were performed on inferred motifs as follows. The motif parameters $\theta$ were first centered and normalized as described above. Each parameter matrix was then unrolled into a $4L \times 1$ vector, where $L = 50$. Standard PCA analysis was then performed on different motif subsets, as shown in panel a.

The inference of motif parameters was performed using a second-generation version of the MPAthic software package[44]. MPAthic enables motif inference using either the ER or IM approaches.

ER inference is the standard way of computing sequence motifs from massively parallel selection experiments[45]. Here, parameter values $\theta^{ER}$ are given by

$$\theta^{ER}_{bl} = \log_2 \frac{f^{selected}_{bl}}{f^{library}_{bl}}, \qquad (2)$$

where $f^{library}_{bl}$ is the fraction of sequences in the initial ARS library that have base $b$ at position $l$, and $f^{selected}_{bl}$ is defined similarly for selected ARS sequences. These fractions were computed using a pseudocount of 1.

IM inference[46] seeks to identify parameters $\theta^{IM}$ that maximize the mutual information between the predicted activity of an assayed ARS sequence and the sample that sequence was observed in. Specifically, one aims to maximize

$$I(\theta) = \sum_{samples} p(sample) \int da\, p(a|sample) \log_2 \frac{p(a|sample)}{p(a)} \qquad (3)$$

where "sample" indicates either the library sample or the selected sample, and $p(a|sample)$ is the distribution of activities assigned to the sequences in that sample by a motif with parameters $\theta$. For a given choice of $\theta$, the distribution $p(a|sample)$ was computed as in Kinney et al.[46]: the activities $a$ in both samples combined were sorted, replaced by their ranks, and binned into 1000 equipopulated bins; for each sample, the distribution of sequence counts across bins was then smoothed using Gaussian kernel having a standard deviation of 20 bins. The marginal probability was subsequently computed as $p(a) = \sum_{samples} p(sample) p(a|sample)$. The optimal values $\theta^{IM}$ were identified using a Metropolis Monte Carlo simulation in which each $\theta$ was assigned relative probability of $2^{NI(\theta)}$, where $N$ is the number of read counts in both the library sample and selected samples combined. Each Monte Carlo run was initiated at random parameter values then carried out for 25,000 steps. Each reported motif resulted from averaging together the end-points of five independent Monte Carlo runs. The present work is the first to show that, as predicted from previous theoretical arguments[47], IM motif inference removes systematic experiment-to-experiment variation that confounds ER motif inference.

**Genome-wide replication origin profile analysis**. Yeast cells were synchronized in G1 with α-factor and were released into medium containing 0.2 mg ml$^{-1}$ pronase E, 0.2 M HU, and 0.5 mM EdU. Cells were collected by centrifugation at 90 min after release into S phase. Genomic DNA was extracted and fragmented[48,49]. EdU-genomic DNA was then biotinylated using the Click reaction and purified using Streptavidin T1 magnetic beads (Invitrogen). Libraries for Illumina sequencing were constructed using TruSeq ChIP Library Preparation Kit (Illumina). Computational analyses of sequencing data are described below (see "Methods", section "Computational analyses of replication origin profile and ChIP-seq data", and "Code availability"). DNA-sequencing data were submitted to the Sequence Read Archive database (see "Data availability").

**Chromatin immunoprecipitation**. The ChIP-seq[50] for Orc1 and Mcm2 were performed with modification as below. About $10^9$ synchronized yeast cells were fixed with 1% formaldehyde for 15 min at room temperature (RT), then quenched with 130 mM glycine for 5 min at RT, harvested by centrifugation, washed twice with TBS (50 mM Tris–HCl pH 7.6, 150 mM NaCl), and flash frozen. Cell pellets were resuspended in 600 μl lysis buffer (50 mM HEPES–KOH pH 7.5, 150 mM NaCl, 1 mM EDTA, 1% Triton X-100, 0.1% Na-Deoxycholate, 0.1% SDS, 1 mM PMSF, protease inhibitor tablet (Roche)), and disrupted by bead beating using multi-tube vortex (Baxter Scientific Products SP Multi-Tube Vortexer S8215-1) for 12–15 × 30 s at maximum setting. Cell extracts were collected and sonicated using Bioruptor (UCD-200, Diagenode) for 38 cycles of pulse for 30 s "ON", 30 s "OFF" at amplitude setting High (H). The extract was centrifuged for 5 min at 20,817 × g. The soluble chromatin was used for IP. Antibody against Mcm2 (mcm228) was preincubated with washed Dynabeads Protein A/G. For each immunoprecipitation, 80 μl antibody-coupled beads was added to soluble chromatin. Samples were incubated overnight at 4 °C with rotation, after which the beads were collected on magnetic stands, and washed three times with 1 ml lysis buffer and once with 1 ml TE, and eluted with 250 μl preheated buffer (50 mM Tris–HCl pH 8.0, 10 mM EDTA, 1% SDS) at 65 °C for 15 min. Immunoprecipitated samples were incubated overnight at 65 °C to reverse crosslink, and treated with 50 μg RNase A at 37 °C for 1 h. 5 μl proteinase K (Roche) was added and incubation was continued at 55 °C for 1 h. Samples were purified using MinElute PCR purification kit (Qiagen). Libraries for Illumina sequencing were constructed using TruSeq ChIP Library Preparation Kit (Illumina). Computational analyses of sequencing data are described below (see "Methods", section "Computational analyses of replication origin profile and ChIP-seq data", and "Code availability"). DNA-sequencing data were submitted to the Sequence Read Archive database (see "Data availability").

**Computational analyses of replication origin profile and ChIP-seq data**. Illumina reads from the genome-wide replication origin profiling and ChIP-seq experiments were mapped to the *S. cerevisiae* S288C genome using BWA, after which pileup files were created using SAMtools (http://www.htslib.org). Pileup counts were then smoothed via convolution with a uniform kernel of width 5000 bp (for replication origin profiles) or 300 bp (for ChIP-seq). To normalize the profiles relative to one another, we computed the number of reads bounding 99.5% of positions within each profile and divided the entire profile by this number.

**Reporting summary**. Further information on research design is available in the Nature Research Reporting Summary linked to this article.

## Data availability
Unique biological reagents used in this study are available upon request to the corresponding author. Raw Illumina reads have been deposited to Sequence Read Archive (SRA) under accession PRJNA595459. Source data are provided with this paper.

## Code availability
Processed data files, analysis scripts, and scripts used for figure generation are available at https://github.com/jbkinney/17_ars.

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

## Acknowledgements

We thank Jennifer Shapp and Kevin Chen for help with experiments. This work was supported by NIH grants R01GM45436 and P01CA13106 to B.S. and R35GM133777 to J. B.K. L.J. is an Investigator of the Howard Hughes Medical Institute. This work was also supported by the Biotechnology and Biological Sciences Research Council (BB/S001387/1, BB/N000323/1) and by the Wellcome Trust (107903/Z/15/Z) to C.S. and NIH grant GM131754 to H.L. The Cold Spring Harbor Laboratory NextGen Sequencing Cancer Center Shared Resource is supported by grant P30 CA045508.

## Author contributions

B.S., J.B.K., Y.H., Y-J.S., L.J., C.S., and H.L. designed the experiments, A.T., W.T.I., and J. B.K. developed the mutual information maximization algorithm. Y.H., J.B.K., and Y.-J.S. performed the experiments. Y.H., J.B.K., and B.S. wrote the paper. All authors analyzed aspects of the data.

## Competing interests

The authors declare no competing interests.
