## [Peer Review File · Nature Communications]

REVIEWER COMMENTS

Reviewer #1 (Remarks to the Author):

I am satisfied with the authors' answers to my comments. They produced novel data and analyses that clarified these questions and rewrote the relevant sections in a much clearer manner. It is somehow surprising that the Orc4 alpha helix deletion mutant is not viable as many species seem to be happy with naturally occurring deletions of this helix. Please specify in the main text exactly which amino-acids were eliminated and if possible discuss the different outcome of the deletion in *S. cerevisiae* and in other species.

Reviewer #2 (Remarks to the Author):

In this revised manuscript "Evolution of DNA Replication Origin Specification and Gene Silencing Mechanisms", Hu et al. have addressed some but not all of the major concerns raised during the initial submission. Thus, as outlined below, some of the conclusions are still not sufficiently supported by experimental data.

1) Abstract: "We show that this Orc4 α -helix mediates the sequence-specificity of origins in *S. cerevisiae*." This sentence overstates the authors' findings as the helix only determines nucleotide preference at two positions in a larger consensus sequence. It does not mediate the recognition of other highly conserved nucleotides in the 11nt A element. Thus, this sentence should be revised.

2) The authors argued in the original submission that altered binding affinities and sequence preferences for ORC binding to DNA lead to altered origin usage in Orc4 mutant strains but had not included any experimental data directly showing this. Such data is still not included in the revision, although the authors have added a genome-wide statistical analysis of origin firing. While the new analysis is consistent with the MPOS findings, both assays do not report directly on ORC sequence specificity and DNA binding affinity, but on origin usage, which is regulated by numerous additional factors. This reviewer considers it essential that DNA binding is directly tested at least for wt and the F485A, Y486A and Y486Q mutants to validate the authors' structural predictions and to support one of the major conclusions of the manuscript, which is that the Orc4 helix contributes to DNA-sequence specificity of ORC binding. As research labs have reopened, this experiment should be doable.

3) Revised Fig. 3: The authors state that they found 2 de novo replication origins in Orc4 mutant strains, which are shown in Fig. 3a and Fig. 3b. Yet, in 3b, the origin designated as completely new is present in the *mrc1* deletion strain, and thus not a novel origin.

4) Extended Data Fig. 11a-j: In the manuscript, the authors refer to orange and green arrow directions in Extended Data Fig. 11 to highlight important results. These arrows are not present and it is unclear what the authors refer to.

5) Page 4: "For example, an active origin in WT and most of the mutant strains was not active in the *orc4*F485I, Y486Q and *orc4*F485A, Y486A strains (Fig. 3c and f). In contrast, many origins that normally do not fire in HU in WT became active in multiple mutants (Fig. 3c, e and h). The firing pattern was mutant dependent." These sentences are confusing as they appear to describe the same data and conclusions that were already explained at the beginning of the paragraph.

6) Page 5: The authors state that the ChIP-Mcm2 results "correspond well to replication origin profiles" in Fig. 3c-h. This statement is not well supported by the data shown in Fig. 3 e-l, as numerous examples in these blow-ups indicate that Mcm2-CHIP signal and origin activity between wt and Orc4 mutant strains poorly correlate. For example, comparing wt and mutant Orc4 profiles (in the presence of *mrc1*):

3e, green arrow: origin activity increases drastically in Orc4 mutant but Mcm2 peak remains unaltered compared to Orc4 wt
3e, blue arrow: origin activity is increased but Mcm2 peak is decreased in Orc4 mutant compared to wt
3e, left red arrow: origin activity is increased but Mcm2 peak remains unaltered in Orc4 mutant compared to wt
3h, right green arrow: origin activity is increased but Mcm2 peak remains mostly unaltered in Orc4 mutant compared to wt
3h, blue arrow: origin activity unchanged but Mcm2 peak decreases in Orc4 mutant compared to wt
3i, red arrow: origin activity increased but Mcm2 peak unaltered in Orc4 mutant compared to wt.

7) Fig. 4. It is unclear what the statistical significance in 4a and above the diagrams in the other panels refers to. What exactly is compared to calculate significance? Are all dinucleotide variants shown in the figure considered in this calculation? Or does it only refer to the AG dinucleotide?

8) Extended Data Fig. 11b-j: The scales of the y-axes are different between the panels. This is misleading and should be corrected.

9) Fig. 1f and Extended Data Fig. 6: The question remains why there is a rather large difference in growth phenotypes in wildtype and some of the mutant strains between liquid culture and plates. For example, the N489W and F485I,Y486Q integration strains have a drastic growth defect at 15 and 30 degrees C on plates in FOA (they almost grow as poorly as the *orc4* null) but have comparably minor defects in liquid culture (1.3-1.4 increase in doubling time). The authors should comment on this behaviour.

10) As pointed out in the original review, the significance of why sequence-specific origin recognition may have evolved together with ORC-Sir4-mediated gene silencing and the loss of RNA interference is unclear. This issue has not been addressed in the revised manuscript. This discussion of this issue is lengthy, taking up ~25% of the paper without providing any experimental data. While interesting, the functional significance and the causality between both observations remain completely unclear. This reviewer strongly recommends that the authors substantially condense this discussion or remove it and include it later in a manuscript describing experimental work related to this topic. Note that reviewer 1 raised similar concerns.

Reviewer #3 (Remarks to the Author):

The additional data included into the revised manuscript further corroborates the original claims by the authors. I agree with the authors that one wouldn't expect to see a lot of "de novo" origin activation, because the mutational changes are relatively subtle. In my mind, this study makes an important contribution to the field, convincingly linking the helix motif in ORC4 to origin sequence recognition. By extension, the missing motif explains the general lack of an origin recognition sequence in metazoa (something that has puzzled the replication community for a long time). I'm also in favor of including the data in Figure 5 that suggest the co-evolution of origin specification and gene silencing. I only have a few suggestions for the following sentences that need editing.

Abstract, line 32: This was accomplished...please delete "analyzed using" and replace with "in conjunction with"

Page 5, line 183: delete "can"

Page 6, line 239: I think "Evolutionally" should be "Evolutionarily"

Page 7, line 276: "specifies" should read "species"

Page 7, lines 300 and 301: "Human" should read "human"

Anja Katrin Bielinsky

Reviewer comments in blue and reply in black.

REVIEWER COMMENTS

Reviewer #1 (Remarks to the Author):

I am satisfied with the authors' answers to my comments. They produced novel data and analyses that clarified these questions and rewrote the relevant sections in a much clearer manner.

It is somehow surprising that the Orc4 alpha helix deletion mutant is not viable as many species seem to be happy with naturally occurring deletions of this helix. Please specify in the main text exactly which amino-acids were eliminated and if possible discuss the different outcome of the deletion in *S. cerevisiae* and in other species.

Thank you for pointing this out. The Orc4 alpha helix deletion is from A481 to Q493. We have specified this in the revised manuscript. Also, we do not think it is surprising that the deletion of the alpha helix is not viable since *S. cerevisiae* evolved to depend on sequence specific origins of DNA replication and deletion of an essential component of that dependence is expected to be lethal. As we discuss in the paper, the *S. cerevisiae* related species acquired the alpha helix and we address the deletion issue when it is mentioned in the paper.

Reviewer #2 (Remarks to the Author):

In this revised manuscript "Evolution of DNA Replication Origin Specification and Gene Silencing Mechanisms", Hu et al. have addressed some but not all of the major concerns raised during the initial submission. Thus, as outlined below, some of the conclusions are still not sufficiently supported by experimental data.

1) Abstract: "We show that this Orc4 α -helix mediates the sequence-specificity of origins in *S. cerevisiae*." This sentence overstates the authors' findings as the helix only determines nucleotide preference at two positions in a larger consensus sequence. It does not mediate the recognition of other highly conserved nucleotides in the 11nt A element. Thus, this sentence should be revised.

We have changed the word "mediates" to "contributes to" in the abstract revised manuscript.

2) The authors argued in the original submission that altered binding affinities and sequence preferences for ORC binding to DNA lead to altered origin usage in Orc4 mutant strains but had not included any experimental data directly showing this. Such data is still not included in the revision, although the authors have added a genome-wide statistical analysis of origin firing.

While the new analysis is consistent with the MPOS findings, both assays do not report directly on ORC sequence specificity and DNA binding affinity, but on origin usage, which is regulated by numerous additional factors. This reviewer considers it essential that DNA binding is directly tested at least for wt and the F485A, Y486A and Y486Q mutants to validate the authors' structural predictions and to support one of the major conclusions of the manuscript, which is that the Orc4 helix contributes to DNA-sequence specificity of ORC binding. As research labs have reopened, this experiment should be doable.

We submitted this paper to Nature in November last year and we received the reviews in February after a long delay. We could not address the reviewer #2's concern in point #2 by doing wet bench experiments because of COVID and since July, CSHL has returned to work at 50% capacity for people at the bench, meaning that the student who will do experiments can only come in half time. Thus, we have not had a chance to make the requested mutants and indeed have not even started to make them, which will take some time. Furthermore, the reviewer #2 is only one of three reviewers requiring us to make mutants and test the binding affinity in vitro.

The reviewer is essentially stating that binding affinity in vitro is the only way to show that a protein has sequence specificity and this is not correct. We agree that binding affinity is one way to show that a protein has sequence specificity, but we disagree that it is the only way to show sequence specificity, which the reviewer implies. Changing sequences in conserved DNA sequence motifs by selection using mutant proteins is another way (long established), and that is what we did using two completely independent assays (MPOS and whole genome origin analysis) and we obtained exactly the same results. This shows definitively that we have changed the preferred sequence of the origins using Orc4 mutants.

Contrary to what the reviewer #2 wrote in his/her review, we never claimed that we demonstrated binding affinity, we only suggested that this is a possibility based on the results. We have, however, changed the paper at multiple points to state that one possibility is that the mutants that change origin sequences change the binding energy or affinity of ORC to DNA. Finally, we never claimed that the Orc4 alpha helix was the only determinant of origin sequence specificity, again contrary to what the review implied. I cannot agree with the reviewer #2 that "some of the conclusions are still not sufficient supported by experimental data". This stated indicates to us that the reviewer is assuming that in vitro DNA binding is the only way to show sequence specificity, and this is simply not true.

3) Revised Fig. 3: The authors state that they found 2 de novo replication origins in Orc4 mutant strains, which are shown in Fig. 3a and Fig. 3b. Yet, in 3b, the origin designated as completely new is present in the *mrc1* deletion strain, and thus not a novel origin.

The reviewer is not correct. First of all, there is no origin predicted in this area from OriDB, thus this is a novel origin location.

Secondly, in Figure 3b, it is obvious that the orange arrow pointed location has origin activity in the *orc4* mutant compared to both WT and *mrc1*Δ, even though the peak is indeed small. There is no obvious peak in the *mrc1*Δ strain at that de novo origin location. It is possible that the reviewer could be looking at a peak just to the right of the orange line in *mrc1*Δ.

4) Extended Data Fig. 11a-j: In the manuscript, the authors refer to orange and green arrow directions in Extended Data Fig. 11 to highlight important results. These arrows are not present

and it is unclear what the authors refer to.

Thank you for pointing this out. This was due to a technical issue that the PDF showed up in black and white when in fact the figure was drawn in color. We now use a PNG version instead.

5) Page 4: "For example, an active origin in WT and most of the mutant strains was not active in the *orc4F485I*, *Y486Q* and *orc4F485A*, *Y486A* strains (Fig. 3c and f). In contrast, many origins that normally do not fire in HU in WT became active in multiple mutants (Fig. 3c, e and h). The firing pattern was mutant dependent." These sentences are confusing as they appear to describe the same data and conclusions that were already explained at the beginning of the paragraph.

Thank you for pointing this out. We have edited manuscript by deleting the duplication in the revised manuscript.

6) Page 5: The authors state that the ChIP-Mcm2 results "correspond well to replication origin profiles" in Fig. 3c-h. This statement is not well supported by the data shown in Fig. 3 e-l, as numerous examples in these blow-ups indicate that Mcm2-CHIP signal and origin activity between wt and *Orc4* mutant strains poorly correlate. For example, comparing wt and mutant *Orc4* profiles (in the presence of *mrc1*):

3e, green arrow: origin activity increases drastically in *Orc4* mutant but Mcm2 peak remains unaltered compared to *Orc4* wt

3e, blue arrow: origin activity is increased but Mcm2 peak is decreased in *Orc4* mutant compared to wt

3e, left red arrow: origin activity is increased but Mcm2 peak remains unaltered in *Orc4* mutant compared to wt

3h, right green arrow: origin activity is increased but Mcm2 peak remains mostly unaltered in *Orc4* mutant compared to wt

3h, blue arrow: origin activity unchanged but Mcm2 peak decreases in *Orc4* mutant compared to wt

3i, red arrow: origin activity increased but Mcm2 peak unaltered in *Orc4* mutant compared to wt.

We have clarified the statement that Mcm2 ChIP "corresponds well with the replication profiles" by stating that "in the *mrc1Δ* strain where all potential DNA replication origins were active, the Mcm2-CHIP profile corresponded well with the DNA replication profile (Fig. 3a and d)" This is what was meant. Note that the *mrc1Δ* EdU profile includes all potentially active origins because it is a checkpoint mutant.

We have also revised the paragraph extensively to point out examples of comparisons between the *orc4* mutant strain and the WT and *mrc1Δ* strain, and indeed there are changes in the Mcm2 ChIP profile in the examples (Fig. 3e-i) that do change along with the replication profile when the *orc4* mutant was compared to WT. This is not universal since we did the experiment looking at early origins (in HU) and thus late origins can still have a peak and yet the origin has not fired.

The revised paragraph is much clearer now.

7) Fig. 4. It is unclear what the statistical significance in 4a and above the diagrams in the other panels refers to. What exactly is compared to calculate significance? Are all dinucleotide variants shown in the figure considered in this calculation? Or does it only refer to the AG

dinucleotide?

We have edited the figure legend accordingly in the revised manuscript to make it clearer.

8) Extended Data Fig. 11b-j: The scales of the y-axes are different between the panels. This is misleading and should be corrected.

Thank you for pointing this out. In the original version submitted to NC, height values were normalized according to the 99.5 percentile value of the peak heights of each mutant in panel b-j for better demonstration of the low correlation in most of the *orc4* mutants. However, the panels k-t were not normalized and they now have been, to make the entire figure consistent. The normalization is described in the revised paper.

The reviewer points out that the y-axes are of a different scale, but this is normalized data comparing the peak heights to the 99.5-percentile of each mutant. We cannot and should not normalize *between* strains, hence the y-axis scale will be different. In any case, the correlation would not be affected by the scaling and therefore we disagree with reviewer# 2's point 8 that the data is misleading and should be corrected on this point. The coefficient of determination values (R^2) are shown on the top of each panel.

9) Fig. 1f and Extended Data Fig. 6: The question remains why there is a rather large difference in growth phenotypes in wildtype and some of the mutant strains between liquid culture and plates. For example, the N489W and F485I,Y486Q integration strains have a drastic growth defect at 15 and 30 degrees C on plates in FOA (they almost grow as poorly as the *orc4* null) but have comparably minor defects in liquid culture (1.3-1.4 increase in doubling time). The authors should comment on this behaviour.

As explained in the last reviewer's response, the strains growing on FOA plates in Extended Data Fig 6 are not the same strains as in Fig. 1f. While the liquid culture in Fig. 1f are the strains where we confirmed the absence of the wild-type *Orc4* on a plasmid with *URA3* selection, the strains in Ext. Data fig.6 are the strains that still contain the wild-type *Orc4* (on a plasmid with *URA3*).

Therefore, the phenotype that we see in Extended Data Fig. 6 is affected by the slower growing phenotype of *orc4* mutants that we see in Fig. 1f together with a loss rate of that wild-type *Orc4* *URA3* plasmid, which could vary in different mutant strains. Thus, the reviewer again misunderstands the different conditions.

10) As pointed out in the original review, the significance of why sequence-specific origin recognition may have evolved together with ORC-Sir4-mediated gene silencing and the loss of RNA interference is unclear. This issue has not been addressed in the revised manuscript. This discussion of this issue is lengthy, taking up ~25% of the paper without providing any experimental data. While interesting, the functional significance and the causality between both observations remain completely unclear. This reviewer strongly recommends that the authors substantially condense this discussion or remove it and include it later in a manuscript describing experimental work related to this topic. Note that reviewer 1 raised similar concerns.

We disagree with the reviewer #2's comment in point #10 about the evolution part and this is his/her opinion, but it is only an opinion. Many others, including the other two reviewers, do find this observation interesting. Thus, we will not agree to take this part out of the paper. Indeed, it is one part of the discussion that makes this evolutionary comparison interesting.

Reviewer #3 (Remarks to the Author):

The additional data included into the revised manuscript further corroborates the original claims by the authors. I agree with the authors that one wouldn't expect to see a lot of "de novo" origin activation, because the mutational changes are relatively subtle. In my mind, this study makes an important contribution to the field, convincingly linking the helix motif in ORC4 to origin sequence recognition. By extension, the missing motif explains the general lack of an origin recognition sequence in metazoa (something that has puzzled the replication community for a long time). I'm also in favor of including the data in Figure 5 that suggest the co-evolution of origin specification and gene silencing. I only have a few suggestions for the following sentences that need editing.

We thank the reviewer for the very thoughtful and positive comments. We also thank the reviewer for the following points.

Abstract, line 32: This was accomplished....please delete "analyzed using" and replace with "in conjunction with". Change was made

Page 5, line 183: delete "can". Change was made

Page 6, line 239: I think "Evolutionally" should be "Evolutionarily". Change was made

Page 7, line 276: "specifies" should read "species". Change was made

Page 7, lines 300 and 301: "Human" should read "human". Change was made and corrected elsewhere.